# Seed Weight as a Covariate in Association and Prediction Studies for Biomass Traits in Maize Seedlings

**DOI:** 10.3390/plants9020275

**Published:** 2020-02-20

**Authors:** Vlatko Galic, Maja Mazur, Andrija Brkic, Josip Brkic, Antun Jambrovic, Zvonimir Zdunic, Domagoj Simic

**Affiliations:** 1Department of Maize Breeding and Genetics, Agricultural Institute Osijek, Južno predgrađe 17, HR31000 Osijek, Croatia; 2Centre of Excellence for Biodiversity and Molecular Plant Breeding (CroP-BioDiv), Svetošimunska cesta 25, HR10000 Zagreb, Croatia

**Keywords:** maize, association mapping, kernel weight, water deficit, genomic prediction

## Abstract

Background: The seedling stage has received little attention in maize breeding to identify genotypes tolerant to water deficit. The aim of this study is to evaluate incorporation of seed weight (expressed as hundred kernel weight, HKW) as a covariate into genomic association and prediction studies for three biomass traits in a panel of elite inbred lines challenged by water withholding at seedling stage. Methods: 109 genotyped-by-sequencing (GBS) elite maize inbreds were phenotyped for HKW and planted in controlled conditions (16/8 day/night, 25 °C, 50% RH, 200 µMol/m^2^/s) in trays filled with soil. Plants in control (C) were watered every two days, while watering was stopped for 10 days in water withholding (WW). Fresh weight (FW), dry weight (DW), and dry matter content (DMC) were measured. Results: Adding HKW as a covariate increased the power of detection of associations in FW and DW by 44% and increased genomic prediction accuracy in C and decreased in WW. Conclusions: Seed weight was effectively incorporated into association studies for biomass traits in maize seedlings, whereas the incorporation into genomic predictions, particularly in water-stressed plants, was not worthwhile.

## 1. Introduction

Developing crop cultivars that can perform well in water-limited environments exacerbated by climate change is an important breeding goal throughout the world. However, drought tolerance is a complex trait demanding a search for additional traits and methods to identify drought tolerant genotypes at different stages of growth. In maize, it is recognized that flowering time is the most critical time for water stress to impact plant performance, although water deficit can damage the field anytime throughout the season by which the seedling stage has received little attention.

Performance traits in maize seedlings are fresh and dry weight, as well as dry matter content of plant biomass, which reliably reflect the effects of water deficit altering the plant’s morpho-physiological status [1]. These traits, however, are assessed by destructive means underscoring the need for some secondary traits which would have higher heritabilities, exhibit enough genetic variability, and are genetically correlated with the performance traits. In this regard, seed size i.e., seed weight expressed as hundred kernel weight (HKW) prior to planting could be an appropriate secondary trait for assessing tolerance for water deficit at the seedling stage. The large seed offers the potential benefit of having larger amounts of available stored energy, while the smaller seed offers the benefit of higher nutrient concentration [2]. However, studies investigating the impact of having larger seed on heterotrophic growth under water-limited conditions are sparse although seed size/weight determines the plant’s early vigor [3] and early growth potential [4].

In recent years, genome-wide association studies (GWAS) were developed into a valuable approach for identifying trait-marker associations overcoming several limitations of biparental quantitative trait loci (QTL) mapping. GWAS provides higher resolution of QTL mapping because it assays a wide range of natural variations resulting from many historical recombination events [5]. Genomic prediction represents a step further from GWAS, and aims to fit all the marker data into a model without prior assessment of the significance of their associations with the phenotype [6]. However, GWAS and genomic prediction methods can be seen as complementary rather than exclusive considering their different aims. The aim of this study is to evaluate the incorporation of seed weight as a covariate into genomic predictions and association studies for three biomass traits in a panel of elite maize inbred lines challenged by water withholding at the end of a seedling stage.

## 2. Results

### 2.1. Genetic Structure and Linkage Disequilibrium

According to the ΔK method for determination of the number of founder populations (K) based on genetic markers, two genetic subgroups were detected in the STRUCTURE analysis (Figure 1a). In alignment with the values of population membership coefficients (Q) of the reference lines B73 and Mo17, the two subgroups were designated Stiff Stalk (SS) and non-Stiff Stalk (NS) for brevity. Principal component (PC) analysis indicated higher diversity in NS compared to the SS subgroup (Figure 1b). The Q matrix for all 109 lines used in this study is available in Appendix A. Linkage disequilibrium decayed across all 10 chromosomes below the arbitrary threshold of *R^2^* < 0.2 at 120 kbp average distance between pairs of markers (Figure 1c). 

### 2.2. Variance Components, Heritabilities, and Responses to Water Withholding

The significant effects of WW treatment were detected for all three examined traits (Table 1). Effects of water withdrawal manifested as significant 41.3% reduction in mean fresh weight (FW), 19.5% reduction in mean dry weight (DW), and 39.7% increase in dry matter content (DMC) relative to control (C). High estimates of genetic variance were observed for all examined traits (Table 1). Interestingly, while the genetic variances of FW and DW have decreased in WW treatment, the genetic variance of DMC increased 4.4 times in WW. A larger estimate of genotype-treatment interaction (72% of σ^2^_G_) was detected for DMC compared to FW and DW in the combined analysis. Very high repeatabilities and heritabilities were estimated for all examined traits. The high repeatability values of HKW, FW, and DMC were obtained because of low estimates of error variances in addition to the high estimated values of σ^2^_G_. Heritability estimates of FW, DW, and DMC across both treatments were expectedly lower because of correction with the genotype-treatment variance. 

Responses of the two genetic subgroups (SS and NS) of inbreds were further analyzed to identify the sources of favorable alleles for responses to water withholding in this stage (Figure 2). Inbreds were split to SS and NS groups according to their respective results from STRUCTURE analysis. Only responses of lines with ancestry coefficient > 0.9 were analyzed in this regard. Similar responses to WW were observed in both groups. Interestingly, lines PHW65 and LH38 from NS pool with related pedigrees showed mild increase in FW values in WW, accompanied with increase in DW. Increase in DW and FW was also observed in three related lines LH132, LH195, and LH205 from the SS pool. The largest relative increase in DMC was observed in line C103 (3.99% in C, 9.84% in WW).

Strong phenotypic correlations were observed for FW and DW between treatments, while correlation of DMC between the treatments was moderate (Table 2, upper triangle). Correlations between FW and DW were strong to very strong within and between C and WW. Weak to moderate correlation was observed between DW and DMC across the treatments, while there were no significant correlations detected for FW and HKW with DMC. Weak to moderate correlations were detected for FW and DW with HKW in both C and WW. Correlations between marker effects of traits calculated using Bayesian ridge regression (BRR) (Table 2, lower triangle) clearly resembled phenotypic correlations, which reflects the high estimates of genetic variances for the traits within the treatments and low error estimates (Table 1). Very low estimates of correlation of marker effects between DMC and FW indicated that there were no shared significantly associated marker loci for these traits. Correlation of marker effects between FW and DW was mildly stronger in WW treatment. Correlation of DMC across the treatments was moderate. Weak correlations between FW, and DW in WW with DMC indicated different mechanisms in genetic regulation of these traits.

### 2.3. Allelic Effects and Candidate Genes

According to the absence of significant correlations between HKW and DMC (Table 2), HKW was not used as covariate in association analysis for this trait. Inspection of quantile-quantile (QQ) plots showed there were no associations crossing the Bonferroni threshold of 5.45 for FW in C and WW and DW in C in GWAS using mixed linear model with covariates population membership (Q), kinship (K) and HKW (MLM + Q + K + HKW), and DMC in WW in MLM + Q+ K without HKW association mapping procedure. Positive deviations from the expected distribution of –log(P) values of associations were observed above value of 3 for DMC in C and above 4 for DW in WW. The total number of loci crossing the arbitrary –log(P) threshold of 4 was 34 (Figure 3), while only 4 loci crossed the calculated Bonferroni threshold of 5.45, single locus for DW in WW and three loci for DMC in C (Figure 3d,e). Two SNPs crossing the arbitrary threshold of 4 were detected for FW in C, and three in WW. For DW, larger number of SNPs crossing the threshold of 4 was detected in C (5) compared to WW (3). Totally 18 loci were detected for DMC in C, and only three in C. Single pleiotropic locus was detected affecting FW in WW and DW in C and WW on chromosome 2, position 17.341 Mbp. This association also had an effect on FW in C, but did not cross the arbitrary threshold of 4 (Figure 3, marked in green, *p* = 0.000325). Interestingly, the effects of this loci were stronger in WW compared to C. An association on chromosome 6, position 101.971 Mbp detected for DMC in C was also detected in WW. Inclusion of HKW as a covariate in FW and DW association analysis yielded 44 percent increase in power of detection as 13 compared to 9 loci were detected (Table 3). The detected loci were distributed over 7 of 10 chromosomes (1, 2, 3, 6, 7, 8, 9). Variance explained by the detected loci (*R^2^*) ranged from 4.35 to 10.06% (Table 3).

### 2.4. Genomic Prediction Models

Genomic prediction models with and without phenotypic covariate HKW were set to elucidate the role of HKW in more precisely accounting for genotypic effects on biomass traits. All assessed predictive abilities were significantly different from zero. Highest predictive ability of the genomic prediction model without covariates was observed for trait DMC in WW treatment (0.62), while the lowest was for FW in C (0.49). Adding HKW as covariate significantly increased the predictive abilities for FW and DW in C. Contrarily, adding HKW as covariate significantly decreased the predictive abilities for FW and DW in WW. Decrease in predictive abilities when HKW was used as covariate was also noted in both C and WW for trait DMC, reflecting the absence of significant correlations between these traits (Figure 4, Table 2). The results of predictive abilities were further corroborated with corresponding values of root mean square error of predictions (RMSEP, Appendix A).

## 3. Discussion

The aim of this study is to evaluate the incorporation of seed weight as a covariate into genomic predictions and association studies for three biomass traits in panel of elite maize inbred lines challenged by water withholding at the seedling stage. HKW is a commonly available information in most breeding programs and it is known to affect early plant growth and development [7].

### 3.1. Genetic Structure

The results of PC analysis in this study were similar to those of Beckett et al. [8] in a study of genetic diversity in ex-PVP germplasm. In their study, STRUCTURE method combined with ΔK population number selection procedure was able to differentiate three independent clusters, namely SS, NS, and Iodent, while in our study Iodent, although visually present, was not empirically different from NS pool because of insufficient number of inbreds with unmixed origin from two putative NS subgroups. Linkage disequilibrium (LD) was higher in current study compared to large diversity panels [8,9] but comparable to recent reports with the medium sized maize diversity panels [10,11]. LD is the utmost factor determining the precision of GWAS. A significant marker-trait association thus means that the causative polymorphism of the trait variation resides within the estimated LD region. Another important implication of LD is that its extent determines the number of genetic markers required to obtain maximum resolution of association analysis [12].

### 3.2. Responses to Water Withholding

Fresh and dry matter accumulation in early vegetative stages of growth is heavily influenced by water deficit, and phenotyping for these traits in diverse germplasm panels under water deficit is one of the first steps in understanding the tolerance to water deficit [13]. Avramova et al. [14] showed that biomass traits such as FW, DW, and DMC are valuable in monitoring responses to water deficit, and can feasibly differentiate even the slight differences in responses of different cultivars. The drought treatment in their study was more severe (14 day WW) compared to 10 days of WW in the current study and the plants were analyzed in different stages compared to this present study (V4 compared to V3) which also changed the scale of responses in all traits. Ge et al. [15] also used these traits for phenotyping two maize inbred lines differing in tolerance to water deficit. It was shown that differences in responses are expected very shortly after the water withholding. Most interestingly, two inbreds from NS pool and two from SS pool in our study considerably increased their DW in WW treatment (>10%). Such responses to WW by increase in DW were reported in literature for drought resistant maize inbred lines [16,17]. The accumulation of DW in WW might have been induced by the accumulation of osmolytes such as proline [18]. 

The choice of mixed modelling approach in variance components analysis was supported by the results of van Eeuwijk et al. [19] that more and more researchers use mixed models to precisely address the GxE variance. The GxT variance in our study is also a form of GxE variance, although with strictly specified conditions of E and changes compared to reference environment (C). The very high repeatabilities of traits examined in this study were expected for experiments set in the controlled conditions addressing the quantitative traits. The low estimates of GxT interaction for FW and DW were accompanied by high levels of plasticity in reactions of examined genotypes for these traits. Plasticity was detected through the correlation analysis in which performances of FW and DW in C were strongly correlated to performances in WW. In contrast, considerable proportion of GxT variance for DMC was accompanied by lower correlation estimates for performance between treatments which indicated the presence of different mechanisms of coping with water deficit, and divergence in responses of different genotypes to stress [20]. The interaction between genotype and watering regime is usually caused by different sensitivities of inbreds to WW treatment [21] thus causing crossover of genotype reactions. These results show that DMC might be more valuable in breeding compared to FW and DW as it allows more sensitivity in discrimination of tolerant and sensitive genotypes to WW. Notably, DMC is calculated only from these two parameters (FW and DW), but it is an inverse to the current water content of plant shoot [14,22]. In field environments, lower heritability values are expected with higher divergence in reactions due to the large number of uncontrolled conditions occurring throughout the growing period. From that perspective, the repeatabilities and heritabilities in controlled conditions are overestimated for quantitative traits.

Line performance for early vigor traits is moderately genetically correlated to general combining ability [23] for that trait, so selection for early biomass accumulation should to certain extent reflect in hybrid performance. Traits FW and DW were moderately correlated to HKW in control which was supported by the findings that early vigor of maize plants is influenced by kernel weight [3,24]. However, the correlations dropped in WW treatment as higher kernel weight appears to offer less advantage when water is sparse [25], as the translocation of nutrients from kernel to shoot is limited. In water deficit genotypic effects on biomass traits become more pronounced, and the selection for early drought tolerance in maize inbred lines should be based on genetic information rather than on kernel weight. However, since the relationship between biomass accumulation and kernel weight is well established [3,4,24] and it represents an a priori available information in many breeding programs. HKW was integrated as covariate into the association analysis of biomass traits FW and DW, while DMC was analyzed with no HKW as covariate due to the absence of significant correlations. The appropriate selection of covariates that are correlated with phenotypes of interest can increase the power of association detection more than two-fold [26]. 

### 3.3. Allelic Effects and Candidate Genes

In correlation analysis between marker effects of traits (Table 2, lower triangle) according to the studies of Ziyomo and Bernardo [27] and Galic et al. [28], it was shown that marker effects for FW in C explain ~56% of variance for FW in WW. The marker effects for DW in C explain ~63% variance for DW in WW, while the marker effects for DMC in C explain only 37% variance of marker effects for DMC in WW. Interestingly, marker effects for FW in WW treatment explain ~68% variance of marker effects for DW in WW treatment. It is expected that FW and DW are highly correlated, as accumulation of osmotically active compounds into the aboveground plant parts is a strategy contributing to increase in both FW and DW in water-limited conditions [14]. The high estimates of correlations between the marker effects for traits between treatments show there is much of the overlap between the allelic effects for these responses, although small number of loci affecting same traits between treatments was detected in GWAS. This is partially explained by the findings that the allelic effects for quantitative traits are highly sensitive to environmental changes that could be attributed to different scenarios [29]. Certain alleles affecting quantitative traits can decrease in effect size in conditions when water is in deficit [30], which could be the reason for the small number of significant allelic effects between treatments despite the high correlations between the marker effects. In addition to that, the limiting factor in detection of loci is still relatively strict threshold of 4 applied in association analysis compared to arbitrary thresholds found in literature.

The GWAS analyses of responses to relevant stresses using phenotypes such as plant’s biomass provide information not only valuable in that particular regard, but also in wider perspective of plant stress responses, as the plant genetic mechanisms of coping with different stresses are similar to certain extent [31]. The search for candidate genes was limited to loci that crossed the Bonferroni threshold. Considering genes located within the estimated 240 kbp window (± estimated linkage disequilibrium block of 120 kbp) from loci crossing Bonferroni threshold, 11 candidate genes were counted for the association on chromosome 1, position 8.775 Mbp, 5 for association affecting FW and DW on chromosome 2, position 17.341 Mb, 1 candidate gene for the association on chromosome 6, position 9.248 Mbp, and 7 candidates for the association on chromosome 6, position 98.450 Mbp. 

The use of phenotypic covariates in association analyses can help gain the statistical power by reduction of noise thus compensating for small sample sizes [26]. Hundred kernel weight is a good covariate in association studies, as it helps differentiate the true genetic from maternal effects, and is as such regularly used in breeding research [32,33]. In our study, not only that the power of association detection was increased, as seen through increase in number of detected loci for FW and DW by 44% (Table 3), but also, some loci have changed positions.

The association on chromosome 1, position 8.775 Mbp, affecting DMC in C is found near the putative cellulose synthase. According to [34,35], this gene is expressed mostly in the topmost leaves of maize plants at three-leaf stage, affecting the cellulose biosynthesis in young plants. This gene plays a key role in the determination of plant architecture, determining the organ size [36]. Cellulose is the building material of cell walls and vascular bundles, and the variations in cellulose synthesis are expected to have effect on dry matter accumulation. Interestingly, the association on chromosome 2, position 17.341 Mbp was found to affect biomass related traits FW and DW in both C and WW treatment, although the effect on FW in C was below the arbitrary threshold of 4, but >3 (Figure 3, marked in green). The associated SNP was located within the position of gene coding for calmodulin-binding protein known to be expressed in SAM and topmost leaf during the three-leaf phase [34,35]. Calcium cell signaling is one of the key plant mechanisms in both tissue specification, as well as sensing of the environmental changes [37], probably causing the increase in effects of this association in WW treatment. Near the association on chromosome 6, position 9.248, affecting the DMC in C, putative gene Zm00001d035195 is found. The expression of this gene is linked to germination and early plant development, and it belongs to LSD family of transcription factors (TFs), probably zinc finger protein (LSD1) [38]. The function of LSD1 might affect the dry matter content in young maize plants, as it was shown to regulate the tissue differentiation in young plants of *Oryza sativa* and promotes callus differentiation [39]. The gene magnesium transporter 2 (*mgt2*) is found in proximity of the association on chromosome 6, position 98.450, affecting DMC in C. The gene *mgt2* controls partitioning of magnesium in developing tissues, and magnesium loading in shoot effects on plant growth are manifold. Growth maintenance requires magnesium translocation [40] first for the utilization in chlorophyll synthesis [41,42], and second for maintenance of osmotic balance, as Mg^2+^ ions are osmotically highly active. Control of vacuolar magnesium contents in developing tissues is a highly important process for maintenance of plant growth affected by many cell-specific processes [43]. 

We speculated that since the relationship between biomass accumulation and kernel weight is established [3,4,24] and it improves the power of association analysis, adding HKW as covariate might also increase the accuracy of genomic predictions. The increase in prediction accuracies in C and decrease in WW confirmed that higher seed weight might facilitate the increase of prediction accuracy given that the water is not in deficit. This was also confirmed in phenotypic studies in maize [44], peanut [45], and soybean [46]. However, in the current study, genomic prediction was shown to be efficient in predicting early biomass accumulation, with accuracies somewhat lower, but comparable to those obtained for biomass accumulation in the later growth stages [47]. Contrarily, Brauner et al. [48] reported comparable prediction accuracies for early vigor when the genomic predictions were performed in lines combined from several genetic pools. Significant differences between means of prediction accuracies obtained with and without HKW as covariate confirmed what was expected after the correlation analysis. The contribution of HKW in biomass accumulation lowers in water deficit which further reflects in lowering of predictive abilities. Adding HKW as a covariate in prediction analysis of DMC decreased the predictive abilities in both C and WW, corroborating the importance of correlation analysis in selection of covariates.

## 4. Conclusions

HKW was weakly to moderately correlated with FW and DW performance in C, while correlations dropped in WW treatment meaning advantages of having larger kernel lower when the water is sparse at this stage. This was corroborated with finding that HKW as a covariate contributes more to predictive abilities of genomic predictions in control than in water withholding. In association analysis for FW and DW adding HKW as a covariate yielded 44% increase in power of detection compared to MLM+Q+K mapping procedure. Four associations were detected passing the Bonferroni threshold. They were mapped in genetically rich regions and their potential value as breeding targets need to be further elucidated. In genomic prediction analysis, adding HKW as covariate has significantly increased the prediction abilities but only in C, while in WW, predictive abilities were lowered. This was in line with the findings from correlation analysis that the advantage of having a larger seed lowers when water is sparse. The use of biomass traits in breeding for tolerance to water withholding in early vegetative stages of growth appears to offer cost-effective, data-rich approach, but more studies are needed linking these traits to other physiological processes thus facilitating the build-up of understanding the processes involved in abiotic stress responses. When analyzing the quantitative traits, use of a priori available covariates is advised. Although their incorporation into the genomic prediction models might not improve the prediction accuracy in stressful conditions, they might serve as a good tool facilitating the more precise dissection of various effects on complex phenotypes.

## 5. Materials and Methods

### 5.1. Plant Material and Experimental Design

The seeds of 154 maize inbred lines with expired PVP certificates were obtained from USA Department of Agriculture NCRPIS (Ames, Iowa). Seeds were planted in growing season 2018 and selfed to obtain sufficient quantity of seeds for experiments. Selfing in sufficient quantity was successful for totally 109 inbred lines. The list of 109 inbred lines with data from patents and repositories is available in Appendix A. 

Experiments were set in growth chamber (25 °C, RH = 50%, 16/8 day/night, 200 µmol/m^2^/s) in trays (510 × 350 × 200 mm). Each tray was filled with 20 l of soil (pH (CaCl_3_) = 5.7, N (NH_4_^+^ + NO_3_^−^) = 70 mg/L, P (P_2_O_5_) = 50 mg/L, K (K_2_O) = 90 mg/L, EC = 40 mS/m) and divided to 15 rows with 7 planting spaces (50 × 35 mm) each. The experiment was set with single water withholding treatment (WW) and control (C) in three replicates. In every tray, 15 genotypes were planted in single row (7 plants). Three trays of each set of 15 genotypes were considered a single replicate. Trays in the growth room were randomly shuffled every day before the lights turned on. Watering regime was optimized in preliminary trials to obtain 50% reduction in fresh weight per plant in WW treatment compared to C. Plants in C were watered continuously across the experiment every two days with spray bottle with 8 mL of tap water per plant. Plants in WW were watered in planting with full dose (8 mL) and continuously thereafter until the fourth day when the watering was stopped. Plant emergence was observed on fourth day, and only plants that emerged on the fourth day were further analyzed (>95%). After that, water was withheld to 14th day (10-day old plantlets) when the aboveground parts of three equally developed plants per genotype in each replicate were harvested and prepared for further analysis. Plants were weighed on a precise four decimal scale to obtain fresh weight (FW) and the 1 g samples were chopped, put in previously weighed 10 mL Falcon tubes and weighed together. Samples were oven-dried at 80 °C, and weighed. Dry weight (DW) was calculated from product of FW and weight of a dried sample/weight of a fresh sample. Dry matter content (DMC) was calculated as (DW/FW) x 100. Additionally, the hundred kernel weight (HKW) was estimated from five self-pollinated ears. Self-pollination was carried out in the growing season of 2018 in a water-managed field. Each inbred line was sown in a two-row plot of 7 m^2^ and ten plants were self-pollinated. Ears of the self-pollinated plants were dried to 10% moisture content prior to shelling. Five well pollinated ears were selected, and their kernels were bulked. Five 100-kernel subsamples were taken from the bulk, and hundred kernel weight (HKW) was estimated by weighing. 

### 5.2. Phenotypic Data Analysis

Variance components of the phenotypic data were analyzed with mixed linear model in *sommer* library [49] using R programming language [50]. Unstructured error variance was assumed. Mixed model was estimated as:$$y_{ijk}=\;\mu +treatment_{i}+genotype_{j}+REP_{k\left(i\right)}+gxt_{ij}+\varepsilon_{ijk}$$
where $y_{ijk}$ represents the k-th observation of the j-th genotype in the i-th treatment, μ represents the grand mean, $treatment_{i}$ represents the effect of the i-th treatment (i=1,2), $genotype_{j}$ represents the genotypic effect (j=1…g), $REP_{k\left(i\right)}$ represents the effect of the k-th replicate (k=1,2,3) within treatment i, $gxt_{ij}$ represents the genotype by treatment interaction, and $\varepsilon_{ijk}$ is the error term. Repeatabilities within the treatments were estimated from variance components as *H^2^* = σ^2^_G_/(σ^2^_G_ + σ^2^_e_/r), while the heritability of each trait was estimated as *H^2^* = σ^2^_G_/(σ^2^_G_ + σ^2^_GxT_/t + σ^2^_e_/tr) where G represents the genotypic effects, GxT is genotype-treatment interaction, e is error, and t and r are the numbers of treatments and replicates respectively. Fisher’s LSD test was performed using the R/*agricolae* package [51] and generalized linear model with treatment, genotype, and replicate main-effects and genotype-treatment interaction using Gaussian family and default link function. 

### 5.3. GBS Data and Filtering

The GBS genotyping was performed at Cornell University [9] according to protocol developed by Elshire et al. [52] and the data were obtained from PANZEA organization [53] repository. The original dataset contained 945,574 SNPs that were annotated, partially imputed, and assigned to chromosomes. Filtering of the low quality SNPs was performed with Tassel software [54] version 5.2.5. Namely, maximum of heterozygotes was set to 0%, minor allele frequency to 2.5%, and maximum of missing data per position to 5%. After filtering there were 229,680 SNPs left. To avoid redundancy of SNPs in similar positions, thinning to 1 SNP per 1000 bp was carried out leaving totally 40,777 high quality markers. 

### 5.4. Genetic Structure Analysis and Linkage Disequilibrium

The genetic structure present in the collection was determined with software STRUCTURE, version 2.3.4 [55]. Random sample of 10,000 SNPs was taken as the input for genetic structure analysis. Analysis was carried out with 10,000 burn-in cycles and 50,000 MCMC runs and population admixture assumed. STRUCTURE analysis was set with ten populations (K = 10) and four runs were carried out for each value of K. The best number of K was 2 and it was chosen according to ΔK method [56] using CLUMPAK software [57]. The germplasm set was divided into groups corresponding to Stiff Stalk (SS) and Non-Stiff Stalk (NS) derived germplasm. Additional run with 50,000 burn-in cycles and 100,000 runs was carried out to assess the population membership estimates of individual genotypes (Q matrix). The Q matrix is available in Appendix A.

The linkage LD decay with physical distance was calculated from all 40,777 filtered markers using Tassel software with window size of 50 bp comprising totally 2,622,025 calculations. The LD decay plot has been constructed in R using the local regression smoothing function *loess*. 

### 5.5. GWAS and Genomic Predictions

Analyses were performed with Tassel software version 5.2.5. Mixed linear modelling approach with population membership (Q) and kinship (K) matrices (MLM+Q+K) was used for GWAS analysis to control for spurious associations and false positives [58]. Hundred kernel weight (HKW) was used as a covariate only for FW and DW according to significant correlation between these traits. The model was calculated according to Bradbury et al. [54] as:$$y=X_{1}\beta_{1}+X_{2}\beta_{2}+X_{3}\beta_{3}+\left(X_{4}\beta_{4}\right)+Zu+\varepsilon$$
where y represents vector of observations, $\beta_{1}$ is a vector of unknown fixed effects containing SNP, $\beta_{2}$ population structure (Q), $\beta_{3}$ replicate factor, and $\left(\beta_{4}\right)$ HKW covariate in MLM+Q+K+HKW model, u is a vector of unknown random additive genetic effects from multiple background QTLs, $X_{1\ldots 4}$ and Z are the design matrices and ε is normally distributed error with zero mean and variance. The variance assumption for u and ε vectors is
$$Var\left(\begin{array}{c} u\\ \varepsilon \end{array}\right)=\left(\begin{array}{cc} G & 0\\ 0 & R \end{array}\right)$$
where $G=\sigma_{a}^{2}K$ with $\sigma_{a}^{2}$ as the unknown additive genetic variance and K is the kinship matrix estimated using identity by state (IBS) method based on all 40,777 filtered markers; R is residual with homogenous variance $R=I\sigma_{\varepsilon }^{2}$ with $\sigma_{\varepsilon }^{2}$ representing the unknown residual variance. The Q matrix for MLM was calculated using all 40,777 SNP markers with principal component analysis with two axes according to the results of ΔK method in STRUCTURE analysis. Correlations between Q_1_ and Q_2_ values assessed by STRUCTURE method and Q_1_ and Q_2_ values assessed by principal component analysis were 0.98 and 0.98 respectively. According to the assessed correlation structure and biological relationship between the traits, HKW was set as phenotypic covariate for FW and DW. Two thresholds for controlling the false detection rate (FDR) were applied. The first threshold was determined according to the Bonferroni correction for α = 0.05 significance level. Namely, the alpha value (α = 0.05) was divided by the effective number of markers ($M_{eff}=14,155$) and the value of 5.4519 was obtained. The value of $M_{eff}$ was calculated according to multiple testing method as implemented in *SimpleM* R script [59,60]. According to the results of Bian and Holland [61] that showed the stable predictive abilities of the loci detected in the range of –log(*P*) thresholds from value of 4 to Bonferroni corrected value in oligogenic and polygenic traits, the second, less-stringent threshold of 4 was applied. Candidate genes were identified through the interface of Maize GDB webpage [62]. The analysis of candidate genes was limited to protein-coding genes within the same linkage disequilibrium block (*R^2^* < 0.2) of 120 kbp around the SNP position crossing the Bonferroni threshold. Physical locations of SNPs are reported according to Maize B73 RefGen_v4 map.

Marker effects were modelled in R/*BGLR* package [63] according to Pérez-Rodríguez et al. [64] with linear covariate as:$$y_{i}=\mu +\overset{p}{\underset{i=1}{\sum }}X_{ij}\beta_{j}+X_{ik}\beta_{k}+\varepsilon$$
where $y_{i}$ is the observed phenotype, $\overset{p}{\underset{i=1}{\sum }}X_{ij}\beta_{j}$ is the sum of marker effects, $X_{ik}$ is a design vector for linear covariate, and $\beta_{k}$ is a scalar of effects for linear covariate. Marker effects were modelled with zero mean and homogenous variance or Gaussian prior density - $p\left(\beta_{j}|\sigma_{\beta }^{2}\right)=N\left(\beta_{j}|0,\sigma_{\beta }^{2}\right)$ with $\sigma_{\beta }^{2}$ representing prior-variance of marker effects corresponding to commonly used ridge regression best linear unbiased prediction (rrBLUP) method in a Bayesian framework e.g., Bayesian ridge regression (BRR). The marker data for calculation of BRR estimates of marker effects were re-coded with Plink software version 1.07 [65] and R interface. K-fold cross validation of genomic prediction models with and without phenotypic covariate was carried out to assess the distribution density of predictive abilities by the use of bootstrapping procedure with 500 random folds of ~20% of phenotypes setting 1000 burn-in cycles and 10000 replicates in the Gibbs sampler per each fold. In cross-validation procedure root mean square error of prediction (RMSEP) was calculated in each bootstrap as $RMSEP=\;\sqrt{\frac{{\left(\sum_{i=1}\mu -y_{i}\right)}^{2}}{n}}$ along with the correlation between predicted and observed values (predictive ability). Differences among predictive abilities and significance of their differences from zero were tested by the means of t-test. The correlations between the marker effects of different traits were calculated in R according to Ziyomo and Bernardo [27] and Galic et al. [28] to estimate the correspondence of small-effect loci governing the different traits even if they fall below the thresholds set for GWAS. Model for estimation of marker effect for correlation analysis was not validated and was set without phenotypic covariates ($y_{i}=\mu +\overset{p}{\underset{i=1}{\sum }}X_{ij}\beta_{j}+\varepsilon$). Calculations were performed in R programming language [50].

## Figures and Tables

**Figure 1 plants-09-00275-f001:**
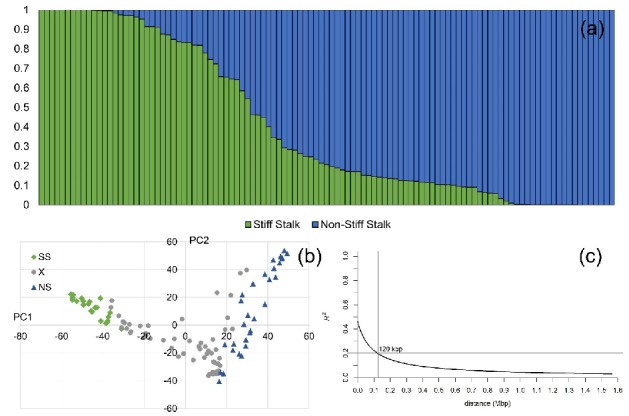
Genetic properties of the 109 genotyped-by-sequencing (GBS) genotyped maize inbred lines: (**a**) Q matrix from STRUCTURE analysis assigning the inbred lines to two groups: Stiff Stalk (SS) and Non-Stiff Stalk (NS); (**b**) results of principal component (PC) analysis using 40777 SNPs—marked green and blue are the lines with Q > 0.9 for SS and NS, respectively; (**c**) linkage disequilibrium decay across all chromosomes.

**Figure 2 plants-09-00275-f002:**
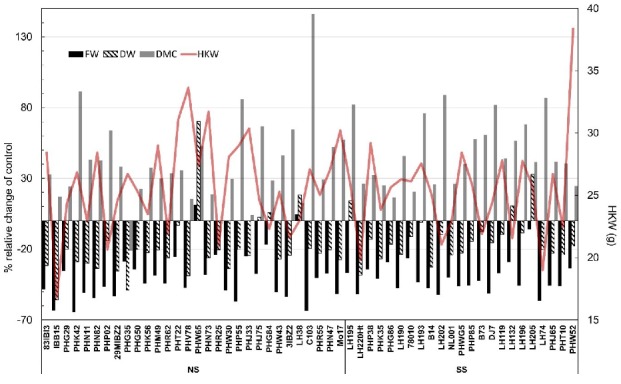
Relative changes in fresh weight (FW), dry weight (DW), and dry matter content (DMC) in water withholding (WW) treatment compared to control (C) of inbred lines with Q > 0.9 belonging to SS or NS genetic group (Figure 1a) along with hundred kernel weight (2nd *y* axis).

**Figure 3 plants-09-00275-f003:**
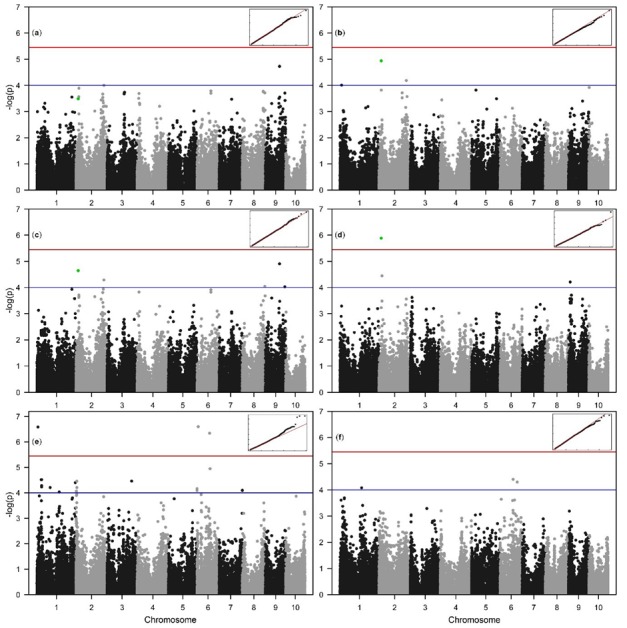
The Manhattan and the respective quantile-quantile (Q-Q) plots of the –log(P) values of the associations from MLM+Q+K+HKW analysis for fresh weight in control (**a**), fresh weight in water withholding treatment (**b**), dry weight in control (**c**), dry weight in water withholding treatment (**d**) and MLM+Q+K for dry matter content in control (**e**) and dry matter content in water withholding treatment (**f**). Blue line represents arbitrary threshold value of 4, while the red line represents the Bonferroni corrected threshold value of 5.45. Marked in green in plots (**a**–**d**) is the SNP S10_139734834 on chromosome 2, associated with Calmodulin binding protein.

**Figure 4 plants-09-00275-f004:**
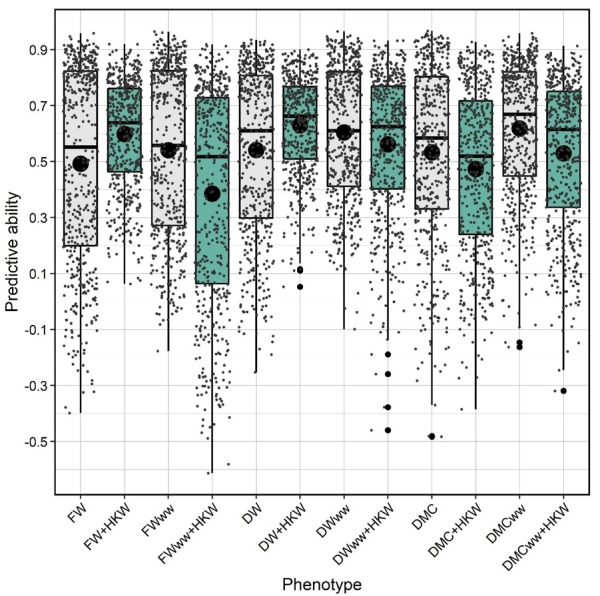
Predictive abilities of genomic prediction models with and without hundred kernel weight (HKW) as a linear covariate for fresh weight (FW), dry weight (DW), and dry matter content (DMC). Dark lines within the boxplots represent the median, and the dark dots represent means of 500 random folds in k-fold cross validation. Gray dots are the predictive abilities from each fold. Differences between mean predictive abilities for each trait with and without covariate are significantly different at α = 0.05.

**Table 1 plants-09-00275-t001:** Means ± standard deviations, ranges, variance components, repeatabilities, and heritabilities for hundred kernel weight (HKW) in grams and biomass traits fresh weight (FW) in grams, dry weight (DW) in milligrams, and dry matter content (DMC) as % of FW in control (C) and water withholding (WW) treatments. The values calculated within treatments are repeatabilities, while the values calculated across treatments represent heritabilities. Repeatability of HKW was calculated for year of seed multiplication (2018).

Trait	Treatment	Mean ± SD ^a^	Range	σ^2^_G_	σ^2^_GxT_	σ^2^_e_	*H^2^*
HKW (g)	–	24.91 ± 4.07	14.68–38.40	14.79	–	1.90	0.96
FW (g)	C	925.8 ± 267.7 ^a^	314.4–1492.2	49868	–	19741	0.88
	WW	540.6 ± 181.8 ^b^	114.1–1002.2	21542	–	10630	0.86
DW (mg)	C	60.29 ± 19.71 ^a^	13.78–106.07	265.00	–	124.10	0.86
	WW	48.57 ± 17.45 ^b^	11.10–87.62	199.00	–	107.40	0.85
DMC (%)	C	6.51 ± 1.05 ^b^	3.99–10.26	0.55	–	0.50	0.77
	WW	9.10 ± 1.86 ^a^	3.96–13.16	2.43	–	1.05	0.87
FW (g)	Combined	733.2 ± 297.4	114.1–1492.2	27431	8295	15135	0.80
DW (mg)	Combined	54.44 ± 19.51	11.10–106.07	210.72	21.11	116.16	0.88
DMC (%)	Combined	7.80 ± 1.98	3.96–13.16	0.86	0.63	0.78	0.66

^a^ different letters denote significant differences between treatments according to LSD test at α = 0.05. ^b^
*H^2^* represents repeatability of HKW and biomass traits within treatments, and heritability in the combined analysis.

**Table 2 plants-09-00275-t002:** Pearson’s correlation coefficients (upper triangle) among fresh weight (FW), dry weight (DW), dry matter content (DMC) between control and water withholding (ww) treatments and with hundred kernel weight (HKW). Lower triangle represents Pearson’s correlations between BRR estimates of marker effects between traits. ** represents significance at α = 0.01, *** represents significance at α = 0.001. All correlations among BRR marker effects are significant at α = 0.001.

	FW	FWww	DW	DWww	DMC	DMCww	HKW
FW	–	0.729 ***	0.921 ***	0.749 ***	0.126	0.117	0.405 ***
FWww	0.792	–	0.632 ***	0.827 ***	−0.001	−0.175	0.309 **
DW	0.918	0.681	–	0.786 ***	0.471 ***	0.336 ***	0.420 ***
DWww	0.825	0.827	0.856	–	0.325 ***	0.381 ***	0.287 **
DMC	0.125	0.053	0.469	0.375	–	0.610 ***	0.102
DMCww	0.252	−0.044	0.479	0.493	0.634	–	0.015
HKW	0.399	0.351	0.367	0.249	−0.043	−0.067	–

**Table 3 plants-09-00275-t003:** SNPs crossing arbitrary threshold of 4 associated with biomass traits fresh weight in grams (FW), dry weight in milligrams (DW), and dry matter content as % of FW (DMC) in control (C) and water withholding (WW) treatments. In bold are the SNPs crossing Bonferroni corrected threshold value for significance at α = 0.05. Shown are the results for MLM + Q + K + HKW analysis for DW and FW and MLM + Q + K for DMC. Results of MLM + Q + K without HKW for FW and DW are available as Appendix A.

Trait	Treatment	Marker	Chr.	Pos.(Mbp)	-log(p)	R^2,a^	SNP	-HKW ^b^
FW	c	S2_212536183	2	219.349	4.003	4.35	C/T	No
FW	c	S9_108404061	9	110.992	4.728	5.32	A/G	No
FW	ww	S1_12465724	1	12.660	4.011	4.61	C/T	No
FW	ww	S10_139734834	2	17.341	4.938	5.67	C/A	Yes
FW	ww	S2_207355968	2	214.210	4.187	5.03	T/C	No
DW	c	S10_139734834	2	17.341	4.643	5.29	C/A	No
DW	c	S2_212536183	2	219.349	4.289	4.75	C/T	No
DW	c	S8_171512464	8	176.755	4.044	4.51	C/T	No
DW	c	S9_108404061	9	110.992	4.906	5.61	A/G	No
DW	c	S9_149744969	9	152.879	4.029	4.92	G/C	No
**DW**	**ww**	**S10_139734834**	**2**	**17.341**	**5.886**	**7.01**	**C/A**	**Yes**
DW	ww	S2_21818202	2	23.110	4.449	5.31	G/A	No
DW	ww	S9_14021178	9	13.709	4.210	5.14	T/C	Yes
**DMC**	**c**	**S1_8741690**	**1**	**8.775**	**6.584**	**9.33**	**C/G**	**–**
DMC	c	S1_34204183	1	34.541	4.516	5.93	C/T	–
DMC	c	S1_37203165	1	37.582	4.232	5.11	A/G	–
DMC	c	S1_37207054	1	37.586	4.518	5.55	A/G	–
DMC	c	S1_37215825	1	37.594	4.284	5.13	A/T	–
DMC	c	S1_101643332	1	103.985	4.208	5.17	C/T	–
DMC	c	S1_173422581	1	175.378	4.03	4.91	T/C	–
DMC	c	S1_295988910	1	301.48	4.401	5.31	G/A	–
DMC	c	S2_2805417	2	2.802	4.068	4.78	T/C	–
DMC	c	S2_6191374	2	6.146	4.456	5.35	C/T	–
DMC	c	S2_7183324	2	7.092	4.207	5.11	G/A	–
DMC	c	S3_189463222	3	192.36	4.461	5.35	C/G	–
DMC	c	S6_127195	6	0.177	4.165	5.18	C/T	–
DMC	c	S6_370986	6	0.392	4.096	4.83	C/T	–
**DMC**	**c**	**S6_8833007**	**6**	**9.248**	**6.596**	**9.07**	**G/C**	**–**
**DMC**	**c**	**S6_95602988**	**6**	**98.45**	**6.342**	**8.41**	**G/C**	**–**
DMC	c	S6_99136681	6	101.971	4.948	6.36	G/A	–
DMC	c	S7_176216182	7	181.799	4.097	4.82	C/A	–
DMC	ww	S1_168415551	1	170.174	4.081	4.89	A/G	–
DMC	ww	S6_99127885	6	101.962	4.403	5.66	A/C	–
DMC	ww	S6_130004982	6	134.089	4.300	5.24	C/T	–

^a^ R^2^ represents percentage of variance explained by the SNP. ^b^ Yes marks if the association was also detected in MLM+Q+K without HKW as covariate (Appendix A).

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
