# Peer review of "Seed Weight as a Covariate in Association and Prediction Studies for Biomass Traits in Maize Seedlings"

_plants, 2020, doi:10.3390/plants9020275_

Round 1
Reviewer 1 Report
General comments:
An Abbreviation List is much needed.
Currently, many parts of the discussion read more like a review than an original paper as the results are not properly discussed. For example the first paragraph of subsection 3.2.
I believe that it makes more sense for the conclusions to come after the discussion and not after the M&M section.
Specific comments:
ABSTRACT
Lines 2-5: The title is too lengthy. Moreover, the authors should perhaps try to make the title reflect the main conclusion/result retrieved from the study?
Line 14: The abstract needs to be improved. The methodology does not correspond to the results presented: currently, it is incomprehensible how did the authors reach the results presented.
Line 14: "Kernel size": Some lack of inconsistency: the authors talk about size and then measure weight?
Line 15: "larger kernel is favorable in seed production" (?)
Do the authors mean as it contributes to yield?
Line 15-17: I do not understand the authors' main objective: in the "context(?) of genotypic data". Please consider rewriting this sentence.
Line 17: It seems that some type of information is missing from the abstract at the beginning of the Methods. It is difficult for the reader to understand the methods used by the author. The elite maize inbreds were genotyped with what?
INTRODUCTION
Line 36: what do the authors mean by fractionating seeds?
Line 37-38: I did not find the information stated in this sentence within the reference used by the authors (reference 5).
Lines 45-61: Too many details on the climate information. Consider reducing this paragraph.
Line 69: How can genome-wide association identify gene function? To the best of my knowledge GWAS per se cannot do this.
Lines 69-72: References are missing in these sentences.
Lines 77-79: I did not understand the meaning of this sentence. How does the genetic variability represent the effects of water deficit on plant morpho-physiological status? please consider revising it.
RESULTS
Line 96: the average distance instead of distance
Lines 126-127: Based in Fig2 the diversity of responses between SS and NS looks similar to me.
Line 128: Stable FW in WW, what does this mean, stable compared to what?
Lines 141: rrBLUP --> are these BLUPs? If so it corresponds to the genetic correlation that should give you an idea if different traits are genetically been controlled by the same regions.
Lines 144-145: I believe that it means that is unlikely that a common genetic basis for the variation of both traits will be detected in your study. It does not necessarily indicate that there is not a "(...) shared genetic background between the traits."
Line 156: No explanation is given on what is BRR except on a legend and on the M&M section. Please verify all the abbreviations used and if there is an explanation to what it stands for at least for the first time it appears in the manuscript.
Line 156: BRR with treatments --> what about the results from the model using no co-variates? did the authors assess the model without using co-variates?
Line 166: where can we find these QQ plots?
Line 167: wasn't the first model (MLM+Q+K+HKW) the model used for genomic prediction and the second model (MLM+Q+K) the model used for association mapping, at least according to what is stated in the M&M section?
Line 171: Bonferroni not Bonferonni
Line 171-173: As it currently stands it seems that the second sentence is the continuation of the first sentence and therefore it does not make sense to have 2 SNPs located in 7 chromosomes. Please rephrase it.
Line 173: to what do these R2 values correspond to?
Line 175-176: if it does pass the threshold defined by the authors it should not consider "affect". You can refer to the P-value of that SNP in the situation where it does not surpasses the threshold.
Line 178-179: again, this methodology is not clear the models used in the association mapping/genomic prediction. as it stands it seems to be all mixed.
Line 181: Usually, a manhattan plot corresponds to the p-value resulted from testing if the phenotypic value between genotypic classes (alleles) is different. Can you please explain why you say that the p-values of the allelic effect(...).
Line 194: At the table header--> Mbp and not MBp. What is R2?
DISCUSSION
Line 197: Please consider starting the discussion by restating the objectives of the work
Lines 23-24: what are the implications of this LD decay. Please discuss more the implications of this.
Line 212: Compared to the 10 days of WW in the current work.
Line 214: What were the directions of responses?
Line 214-217: how do the results from this work compare to Sekhon et al.? If I understand correctly in this work only aerial part was assessed and not different tissues so how can you compare the Sekhon results to your own?
Line 220-221: Why is this information on the reflectance indices important to your work?
Lines 223-225: How do the increases in osmolytes and increased activities of antioxidant pathways lead to an increase in the dry weight?
Lines 225-226: I do not understand this sentence: what is the "genetic diversity confounded"? which "diversity of reactions" do the authors are referring to?
Lines 227-232: so what were the main conclusions of the GxT results and how do they compare to what was previously known?
Lines 233-242: join this paragraph with the previous one.
Lines 236-239: Given that dry matter content is only calculated based on fresh and dry weight how can you explain this?
MATERIALS AND METHODS
Line 321: The authors got 105 plants per tray, is that correct? and 15 different genotypes per tray? Where were the replicates placed?
Line 321: I do not understand the use of these boxes: the soil was not placed inside the traits? Where do these boxes come to play? Or the soil was placed inside the boxes?
Line 323-324: this experimental design can bias the results since all the plants from the same genotype seem to be sown in the same row.
Line 325: Continuous watering across the experiment or continuous water flow?
Line 326: 8 mL per day?
Line 329-331: Why did the authors decide not to evaluate the root (below the ground)?
Line 333-334: something does not seem right in this sentence. please revise it.
Line 335-337: The experimental design and field conditions are missing for such an important trait. Year of trial? How do you explain using only a year of field trials? are the samples used in the field trial as the ones grown in controlled conditions?
Line 336: Five random samples per genotype? per replicate?
Line 339: A description of the model used is missing: random terms? fixed terms? Mixed models were used to obtain specifically what estimates?
Line 334-345: Which post hoc test? For GLM which terms were used in the model, link function, distribution?
Line 363: To what corresponds to the Q matrix?
Line 365: How many markers were used for LD decay calculations?
Line 368: This is a result.
Line 370: Please define your model.
Line 370: Why did the authors decide to use the MLM+Q+K model? did the authors test other models?
Lines 372-375: What is the difference between the Q matrix and the PC?
Line 375: Plot values --> now I am really confused: what plots do the authors refer to? Wasn't the experiment done in trays/boxes?
Line 376-377: what do the authors mean with expected associations between traits? shouldn't the associations be between markers and traits or am I missing something? Please explain.
Line 377: Phenotypic co-variate --> once again the model used for association mapping is missing to understand exactly the methodology followed by the authors. As it stands in the present the M&M section is not sufficient.
Lines 377-378: which two thresholds? or a threshold of 2?
Line 390: The correlations were calculated using which software?
Line 396: So the rrBLUP model was used to estimate BLUPS. And these were used for genomic prediction? This is not clear.
Line 400: Fixed linear covariate--> I have doubts about the validity of using the Treatment as a linear co-variable. Do you have any reference to add here to validate this option?
Author Response
Dear reviewer,
the responses are in the file attached.

Reviewer 2 Report
See the attached file.

Author Response
Dear reviewer,
the responses are in the file attached.
Sincerely

Round 2
Reviewer 1 Report
The authors improved considerably the manuscript and I only have some some minor comments to address (the lines refer to the document with the track-changes on).
Line 38 - Consider not using abbreviations to define your keywords
Line 126 - "trait-marker associations" or "traits-molecular marker associations" (since polymorphism can be misleading)
Author Response
Dear reviewer,
thank you for your time and effort in reviewing our manuscript. We hope that the manuscript is now acceptable for publication in the present form.
The abbreviations in keywords were replaced by fully written synonyms (now line 25) and the “trait-polymorphism” was replaced by trait-marker (now line 52), along with several corrections of punctuation and spaces after the acceptance of changes from previous round of review.
Sincerely,
in the name of all co-authors,
Vlatko Galić